# UniWav: Towards Unified Pre-training for Speech Representation Learning and Generation

**Alexander H. Liu**[1†], **Sang-gil Lee**[2], **Chao-Han Huck Yang**[2], **Yuan Gong**[1‡],
**Yu-Chiang Frank Wang**[2], **James R. Glass**[1], **Rafael Valle**[2], **Bryan Catanzaro**[2]
[1]MIT CSAIL, [2]NVIDIA
alexhliu@mit.edu

## Abstract

Pre-training and representation learning have been playing an increasingly important role in modern speech processing. Nevertheless, different applications have been relying on different foundation models, since predominant pre-training techniques are either designed for discriminative tasks or generative tasks. In this work, we make the first attempt at building a unified pre-training framework for both types of tasks in speech. We show that with the appropriate design choices for pre-training, one can jointly learn a representation encoder and generative audio decoder that can be applied to both types of tasks. We propose UniWav, an encoder-decoder framework designed to unify pre-training representation learning and generative tasks. On speech recognition, text-to-speech, and speech tokenization, UniWav achieves comparable performance to different existing foundation models, each trained on a specific task. Our findings suggest that a single general-purpose foundation model for speech can be built to replace different foundation models, reducing the overhead and cost of pre-training. Audio demo page: https://alexander-h-liu.github.io/uniwav-demo.github.io/

## 1 Introduction

Representation learning and generative modeling have seen rapid growth in speech processing in recent years. Researchers have been consistently setting new records on discriminative tasks with representation learning (Baevski et al., 2020b; Hsu et al., 2021; Baevski et al., 2022; Liu et al., 2024). Existing speech representations focus on excelling at a specific task. Mainstream pre-training methods (Schneider et al., 2019; Baevski et al., 2020a;b; Hsu et al., 2021), also known as self-supervised learning, learn speech representation from massive unlabeled speech. While there are many different prior works in the field, they all shared the same spirit in masked audio modeling, i.e., matching the output extracted from masked audio to that from unmasked audio. These representations tend to capture the phonetic or semantic content of speech, leaving out acoustic information such as speaker's timbre and the acoustic environment. Consequentially, self-supervised representations have dominated classification and recognition tasks (Yang et al., 2021) in speech but not for generation tasks (Tsai et al., 2022). In fact, recent research (Wang et al., 2023) argues that self-supervised representations (Hsu et al., 2021) are sub-optimal for speech synthesis.

On the other hand, generative models have also gained more attention in recent years. State-of-the-art models are capable of generating realistic speech with arbitrary voice and text (Wang et al., 2023; Le et al., 2023), also supporting a wild range to generation tasks beyond text-to-speech (Wang et al., 2024). Similar to representation learning, pre-training methods for speech generative models have also been developed (Liu et al., 2023; Vyas et al., 2023) based on masked audio modeling. These models also are pre-trained on unlabeled data and can be easily transferred to different downstream tasks with minimal supervision.

---

[†]Work done during an internship at Nvidia. [‡] Work done at MIT, now with xAI Corp.

Interestingly, while the two families of models are similar in many ways — e.g., training transformers, applying masked audio modeling, pre-training with unlabeled data, etc. — they are normally designed and trained separately.

Ideally, speech representation learning and generation should be complementary, as representation learning can be used to guide generation, and generation can be used to ensure that the learned representation encodes sufficient information. However, combining them is challenging due to conflicting methods and arguably opposing requirements for each task (Baevski et al., 2022). For example, while speech recognition tasks can benefit from a representation that is invariant to speaker and environment (Schneider et al., 2019; Hsu et al., 2021), speech synthesis tasks require a representation that provides information about speaker and environment (Wang et al., 2023; Zhang et al., 2023b).

Nevertheless, there are successful examples on text (Devlin et al., 2018; Radford et al., 2019) and image (Li et al., 2023; Xiang et al., 2023; Chen et al., 2024) that described unified foundation models for representation learning and generation. These prior works showed that a unified pre-training framework for representation learning and generation can reduce the overhead of training, while accommodating both discriminative and generative downstream applications with strong performance.

Our proposed framework, UniWav, is the first to present a unified pre-training framework for speech that is efficient on both discriminative and generative tasks. It is an encoder-decoder framework, integrating a representation encoder and a Flow Matching (Lipman et al., 2023) decoder conditioning on the learned representations. The representation encoder is guided simultaneously by self-distillation (Baevski et al., 2022; Liu et al., 2024) and the generative objective through the decoder. By training the encoder and decoder jointly in an end-to-end manner from scratch, we build the first foundation model that excels in both speech recognition and generation. In our experiments, we fine-tune the pre-trained model for speech recognition and in-context text-to-speech synthesis, achieving results comparable to state-of-the-art methods *in each task*. Additionally, we adapt UniWav for speech tokenization, demonstrating the significant benefits of joint representation and generation learning by achieving high audio quality at a low bitrate. Finally, we provide observations and insights on unified pre-training through an ablation study and analysis, aiming to illuminate the path toward a single foundation model for general speech processing.

Our contribution can be highlighted as follows:

- We introduce, UniWav, the first unified pre-training framework for speech representation learning and generation.
- On speech recognition and in-context text-to-speech generation, we show UniWav can compete with different foundation models within each task through fine-tuning.
- By introducing a discrete bottleneck between the encoder and decoder, UniWav achieves low-bitrate speech tokenization and high-quality resynthesis, outperforming existing methods.

In summary, UniWav marks the first step toward unified pre-training for both representation learning and generation, laying the foundation for more versatile pre-trained models in speech processing.

## 2 METHOD

Figure 1 provides an overview of UniWav, an encoder-decoder framework designed to unify pre-training for representation learning and generative tasks. The model takes the surface feature (e.g., Encodec (Défossez et al., 2022) latent or mel spectrogram) of speech as input. Through masked audio modeling, the model jointly learns to encode input into latent representations and generate speech. Below, we detail encoder design in §2.1, decoder in §2.2, and summarize the pre-training framework in §2.3.

### 2.1 REPRESENTATION ENCODER

The key to learning representations from unlabeled data is to derive good learning targets from the input itself, commonly referred to as self-supervised learning. While there are different self-supervised learning methods in speech (see related works §4), we adopt self-distillation and online clustering from DinoSR (Liu et al., 2024) in this work.

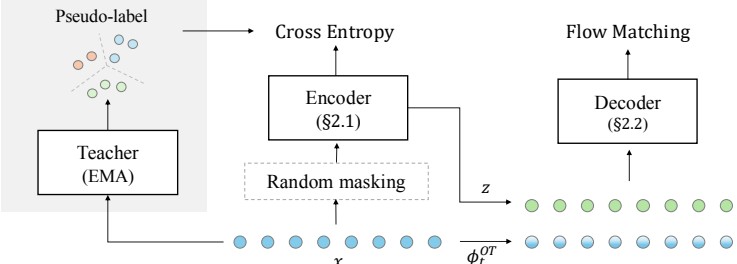

Figure 1: An overview of UniWav. The encoder is trained with masked audio modeling and pseudo-label obtained through a teacher model. The teacher model is the exponential moving average (EMA) of the encoder. The decoder is trained with Flow Matching conditioned on $z$ the weighted sum of representations of different encoder layers. All modules are trained jointly from scratch.

Given the sequence of input audio frames $x = (\mathbf{x}_1, \mathbf{x}_2, \ldots, \mathbf{x}_L)$ where $\mathbf{x}_i \in \mathbb{R}^{d_x}$ is the input surface feature frame of speech, we randomly mask a subset of the frames, denoted $M$, and feed them into the Transformer encoder (Vaswani et al., 2017) to derive output latent embeddings $z = (\mathbf{z}_1, \mathbf{z}_2, ..., \mathbf{z}_L)$ where $\mathbf{z}_i \in \mathbb{R}^d$. The goal of the encoder is to predict the pseudo label $y_i$ provided by a teacher model by maximizing

$$\frac{1}{|M|} \sum_{i:\mathbf{x}_i \in M} \log p_\phi(y_i|\mathbf{z}_i), \tag{1}$$

where $\phi$ is the pseudo-label prediction head (linear layer followed by softmax activatiton).

The teacher model for deriving pseudo-label is the exponential moving average (EMA) of the encoder itself, i.e., $\theta_{\text{teacher}} \longleftarrow \gamma_{\text{teacher}} \theta_{\text{teacher}} + (1 - \gamma_{\text{teacher}}) \theta_{\text{encoder}}$ where $\gamma_{\text{teacher}}$ is the decay factor. While both networks are randomly initialized and set to be identical at the beginning of training, only the encoder takes masked audio as input. In contrast, the teacher model takes complete audio (without masking) as the input to derive the pseudo-label

$$y_i = \operatorname*{argmin}_{v \in V} \|\tilde{\mathbf{z}}_i - \mathbf{e}_v\|_2 \tag{2}$$

for input frame $\mathbf{x}_i$, where $\tilde{\mathbf{z}}_i$ is the output of the teacher model and $\mathbf{e}_v$ is a codeword from a codebook of size $V$. All codewords within the codebook are also updated by taking the EMA of the teacher output embeddings that share the same pseudo-label, i.e.,

$$\begin{aligned}
\mathbf{s}_v &\longleftarrow \gamma_{\text{code}} \, \mathbf{s}_v + (1 - \gamma_{\text{code}}) \sum_{i:y_i=v} \tilde{\mathbf{z}}_i, \\
n_v &\longleftarrow \gamma_{\text{code}} \, n_v + (1 - \gamma_{\text{code}}) \sum_{i:y_i=v} 1, \\
\mathbf{e}_v &\longleftarrow \frac{\mathbf{s}_v}{n_v},
\end{aligned} \tag{3}$$

where $\gamma_{\text{code}}$ is the decay factor, the cluster sum $\mathbf{s}_v \in \mathbb{R}^d$ is randomly initialized, and the cluster size $n_v$ is initialized to 1.

Following DinoSR, we consider pseudo-labels from all the top K layers of the teacher model by repeating Eq.(1),(2), and (3) for each layer. The complete loss function of the encoder is therefore

$$\mathcal{L}_{\text{encoder}} = -\frac{1}{|K| |M|} \sum_{i:\mathbf{x}_i \in M} \sum_{k \in K} \log p_{\phi_k}(y_i^{(k)}|\mathbf{z}_i), \tag{4}$$

where $K$ is the set of indices of the teacher model layer to be considered, each with an independent codebook. Note that the prediction is made at the final layer, using $\mathbf{z}_i$, of the encoder regardless of the target layer $k$.

## 2.2 FLOW MATCHING DECODER

We choose to train our decoder with Flow Matching (Lipman et al., 2023), which has been shown to be highly effective and generalizable in speech generation tasks (Le et al., 2023; Liu et al., 2023; Kim et al., 2024; Mehta et al., 2024).

Formally, Flow Matching considers the problem of learning to generate samples $x_1$[1] of an unknown data distribution $q(x)$ by predicting the *flow* of $x$ along the path between a prior distribution $p_0$ and the target distribution $p_1 \approx q$. The flow is defined using ordinary differential equation (ODE):

$$\frac{d}{dt}\phi_t(x) = v_t(\phi_t(x)); \quad \phi_0(x) = x; \tag{5}$$

with the time-dependent vector field $v_t : [0,1] \times \mathbb{R}^d \to \mathbb{R}^d$. Through the change of variable formula, we can generate the distribution $p_t$ at any timestep $t$ along the path, allowing us to sample $p_1$ by solving Eq.(5) iteratively from $t = 0$ to $t = 1$.

Under the framework, the goal of our decoder is to predict the time-dependent vector field $v_t(x; \theta_{\text{decoder}})$ that matches the target vector field $u_t$ that generates the path. While the framework is simple, naïve Flow Matching - regressing decoder output to $u_t$ - is intractable since the target vector field $u_t$ is generally unknown in practice. Fortunately, Lipman et al. (2023) showed that we can leverage real data sample $x_1$ to construct the conditional flow of $x$ with tractable $p_t$ and $u_t$. Specifically, the Optimal Transport (OT) conditional path is constructed by assuming the mean $\mu_t(x) = tx_1$ and standard deviation $\sigma_t(x) = 1 - (1 - \sigma_{\min})t$ to change linearly in time, with a sufficiently small $\sigma_{\min}$ such that $p_1(x|x_1)$ is centered around $x_1$. This results in the OT conditional flow

$$\phi_t^{OT}(x) = (1 - (1 - \sigma_{\min})t)x_0 + tx_1 \tag{6}$$

with tractable $p_t(x|x_1) = \mathcal{N}(x \mid \mu_t(x_1), \sigma_t(x_1)^2 I)$ and $u_t(x|x_1) = \frac{(x_1 - (1 - \sigma_{\min})x)}{(1 - (1 - \sigma_{\min})t)}$, deriving the Conditional Flow Matching (CFM) objective

$$\mathbb{E}_{t, q(x_1), p_0(x_0)} \left\| v_t(\phi_t^{OT}(x_0); \theta_{\text{decoder}}) - \left( x_1 - (1 - \sigma_{\min})x_0 \right) \right\|^2, \tag{7}$$

where $p_0$ is the standard normal distribution and $t$ is sampled uniformly from $[0,1]$. It is worth mentioning that Lipman et al. (2023) proved the CFM objective and the naïve Flow Matching share identical gradient w.r.t. the model, therefore Eq.(5) can be solved with the model trained with CFM.

We additionally condition the decoder on the weighted sum of the encoder layer features $z$, yielding the complete loss function for the decoder:

$$\mathcal{L}_{\text{decoder}} = \mathbb{E}_{t, q(x_1), p_0(x_0)} \left\| v_t(\phi_t^{OT}(x_0), z; \theta_{\text{decoder}}) - \left( x_1 - (1 - \sigma_{\min})x_0 \right) \right\|^2. \tag{8}$$

Similar to the representation encoder, the generative decoder is also implemented with a Transformer encoder with $d$-dimensional latent. The input of the decoder is

$$W_0 v_t(\phi_t^{OT}(x_0)) + z, \quad \text{where} \quad z = \sum_i W_i z^{(i)}, \tag{9}$$

$W_0 \in \mathbb{R}^{d_x \times d}$ is the noisy data projection layer, and $W_i \in \mathbb{R}^{d \times d}$ is the linear projection layer for the representation $z^{(i)}$ from the $i$-th layer of the encoder. In other words, the decoder takes the noise-injected audio feature $x$ and a learnable combination of representations from all encoder layers $z$ as input.

## 2.3 UniWav Framework

Combining the encoder objective Eq.(4) and the decoder objective Eq.(8), UniWav is trained from scratch in an end-to-end manner with the overall objective

$$\mathcal{L}_{\text{encoder}} + \lambda \mathcal{L}_{\text{decoder}}, \tag{10}$$

where $\lambda$ is the weighting scalar. In practice, we found downstream performance less sensitive to $\lambda$ but a sufficiently small value stabilizes pre-training. We use $\lambda = 0.25$ throughout this work. Note that for both encoder and decoder, we only compute loss at the masked positions.

Unlike prior works that rely on iteratively learning representation from teacher models (Hsu et al., 2021; Chen et al., 2022a), adopting self-distillation and online clustering allows training the encoder

---

[1]In this subsection, we denote the sequence of audio frames as $x_1$, i.e., $x_1 := (\mathbf{x}_1, \mathbf{x}_2, \dots, \mathbf{x}_L)$, to follow the convention in the literature.

from scratch together with the decoder. In practice, we also tested self-distillation without clustering (Baevski et al., 2022) and distilling from a pre-trained HuBERT (Hsu et al., 2021), but found the overall encoder-decoder training less stable. For the decoder, we tested reconstructing discrete code from Encodec (Défossez et al., 2022) and found Flow Matching better. We leave exploring more combinations of the encoder and decoder as an important future work.

After pre-training, the encoder is a stand-alone module that can be used to extract representations similar to those of prior works. We evaluate the representation encoder with speech recognition in §3.2 and speech tokenization in §3.4. Additional representation analysis is presented in §3.5. For generation, the decoder can be fine-tuned using desired conditions such as text and audio prompt (§3.3) and discretized representations (§3.4).

## 3 EXPERIMENTS

### 3.1 PRE-TRAINING SETUP

**Model** Both the encoder and the decoder follow the Transformer architecture (Vaswani et al., 2017), with 16 attention heads, hidden size $d = 1024$, and feed-forward networks with 4096 dimensions. For the encoder, positional information is encoded using convolutional positional embedding (Baevski et al., 2020a). We add skip connections between decoder layers to mimic the U-Net (Ronneberger et al., 2015) architecture, following prior works (Le et al., 2023; Liu et al., 2023). Both the encoder and the decoder use the ALiBi self-attention bias (Press et al., 2021). We stack 24 layers for the encoder and 12 layers for the decoder. To condition the Flow Matching decoder on ODE timestep $t$, a sinusoidal embedding of $t$ is prepended to the input sequence of the decoder. The total number of learnable parameters is around 500M, where the encoder accounts for 60% of the size. Studies on the size of the encoder and the decoder are presented in §3.5.

**Data & Input Feature** UniWav is trained on LibriLight (Kahn et al., 2019), an audiobook corpus with around 60k hours of untranscribed English speech sampled at 16kHz. We extract the pre-quantized latent feature from a 16kHz EnCodec (Défossez et al., 2022) encoder, pre-trained on the same dataset, to serve as the input for our model. The input feature is comprised of 50Hz 128-dimensional vectors, normalized to zero mean and unit variance overall. For generation tasks, the features sampled by the decoder can be projected back to waveform using the EnCodec decoder. Choice of surface feature is studied in §A.1 by replacing EnCodec latent with mel spectrogram.

**Hyper-parameters & Training details** For the encoder teacher model, $\gamma_{\text{teacher}}$ is set to increase from 0.9997 to 1.0 in the first 400k updates, and $\gamma_{\text{code}}$ is set to 0.9. The top $K = 10$ layers of the teacher model are considered a learning target for the encoder, and the representation from each layer is clustered to a codebook of size $V = 256$, following Liu et al. (2024). For masking, each input frame has an 8% chance of being replaced with a learnable mask embedding. We mask 10 consecutive frames if a frame is sampled to be masked. We set the decoder loss weight $\lambda$ to 0.25. UniWav is trained using Adam optimizer (Kingma & Ba, 2014) with a cosine learning rate schedule peaking at 2e-4 for a total of 600k updates, including 10k warmup steps. The batch size is 312.5 seconds per GPU, and samples are randomly cropped to cap at 20 seconds. The model is trained with bf16 and gradient clipping of 1.0. Pre-training is done on 16 H100 GPUs taking around 9 days.

### 3.2 SPEECH RECOGNITION

**Setup** We follow the prior works in self-supervised representation learning (Baevski et al., 2020a; Hsu et al., 2021; Chen et al., 2022a; Baevski et al., 2022) to evaluate the UniWav encoder on speech recognition. A linear layer for letter prediction is attached to the final output layer of encoder, and the whole encoder is fine-tuned with CTC (Graves et al., 2006). In alignment with the prior works, we fine-tune our encoder using either 960 hours of transcribed speech from LibriSpeech (Panayotov et al., 2015) or 100 hours from its `train-clean-100` subset. We use the open-sourced fine-tuning and decoding script from Baevski et al. (2020a), and perform a hyperparameter search over learning rate, mask probability, mask length, and input channel dropout probability. For decoding, the LibriSpeech 4-gram language model is used.

Table 1: Comparison of speech foundation models. For speech recognition, we report word error rate (WER) on the `test-other` subset of the Librispeech (Panayotov et al., 2015) dataset when fine-tuning with 960 hours or 100 hours of labeled data. For speech synthesis, we report ASR-measured WER (ASR-WER) and speaker similarity (Sim.) for in-context text-to-speech.

| | Pre-training | Speech Recognition | | Speech Synthesis | | |
|---|---|---|---|---|---|---|
| | data (hr) | 100hr | 960hr | data | ASR-WER | Sim. |
| Self-supervised Representation Models | | | | | | |
| HuBERT (Hsu et al., 2021) | 60k | 4.5 | 3.7 | - | - | - |
| WavLM (Chen et al., 2022a) | 94k | 4.6 | - | - | - | - |
| data2vec (Baevski et al., 2022) | 60k | 4.6 | 3.7 | - | - | - |
| data2vec 2.0 (Baevski et al., 2023) | 60k | 4.3 | 3.5 | - | - | - |
| Generative Models | | | | | | |
| VALL-E (Wang et al., 2023) | 0 | - | - | 60k | 3.8 | 0.508 |
| Voicebox[†] (Le et al., 2023) | 0 | - | - | 60k | 2.3 | 0.616 |
| SpeechFlow[†] (Liu et al., 2023) | 60k | - | - | 960 | 2.4 | 0.629 |
| UniWav[†] w/o pre-training | 0 | 17.5 | 5.8 | 960 | 2.6 | 0.532 |
| UniWav[†] | 60k | 4.8 | 4.0 | 960 | 2.5 | 0.635 |

[†] Inference with time-aligned phone sequence using force alignment (McAuliffe et al., 2017); 32 function evaluations.

**Results** We report word error rate (WER) on the `test-other` subset in Table 1. The clear gap between random initialized and pre-trained weights demonstrated the effectiveness of pre-training. Comparing UniWav with prior works on representation learning, we note that our model falls short of state-to-the-art by around 0.5% WER. Nevertheless, we note that prior works in this direction focused exclusively on discriminative tasks, and none of the prior works supported generation tasks. The speech recognition experiment confirmed that the encoder of UniWav can effectively function as the backbone for discriminative tasks, even when pre-trained with the additional demands of a generative objective.

## 3.3 SPEECH SYNTHESIS

**Setup** Next, we evaluate the speech generation ability of UniWav against existing foundation models on in-context text-to-speech (Wang et al., 2023; Le et al., 2023; Liu et al., 2023). We fine-tune UniWav on LibriSpeech to synthesize speech, conditioned on text and audio prompts that provide speaker information. For text encoding, we use input-frame-aligned phone labels with position postfixes (Le et al., 2023). Phone embeddings are randomly initialized and added to the model input. For audio prompts, we directly use the encoder representation ($z$ in Eq.(8)), where 70% to 100% of the sequence is randomly masked with a single mask. We also drop both audio prompt and text condition with a 20% chance for classifier-free guidance (CFG; Dhariwal & Nichol, 2021; Le et al., 2023).

We fine-tune our model with 1e-5 learning rate for 150k steps with logit-normal time sampling (Esser et al., 2024) on 8 A100 GPUs. Note that since our goal is to evaluate acoustic modeling, we use Montreal Force Aligner (McAuliffe et al., 2017) to obtain alignment for both training and evaluation (and apply the same for prior works that require alignment). For solving the initial value problems in Eq.(5), we use the midpoint method from `torchdiffeq` (Chen, 2018). With a step size of 0.0625 and CFG $\alpha = 1.9$, the number of function evaluations (NFEs) is 32 per sample.

For evaluation, we adapt the protocol introduced by Wang et al. (2023) to perform speaker-conditioned text-to-speech on the `test-clean` subset with 3-second enrollment. We follow Liu et al. (2023) to use fine-tuned HuBERT-L (Hsu et al., 2021) to measure word error rate for audio intelligibility (ASR-WER) and WavLM-TDCNN speaker embedding model (Chen et al., 2022a) to measure speaker similarity (Sim) for speaker consistency between the audio prompt and the result.

**Results** We compared UniWav against prior works using the same setup in Table 1. Compared to Vall-E (Wang et al., 2023) and Voicebox (Le et al., 2023) that relied on speech data with transcriptions, UniWav achieved better or similar results with significantly less data. Compared to the generative pre-training method SpeechFlow (Liu et al., 2023), our method shows equal performance in fine-tuning. This finding is important because, unlike SpeechFlow, UniWav can also be applied for discriminative

Table 2: Speech tokenization and resynthesis results. All models have a frame rate of 50Hz. Methods sharing the same bitrate are highlighted with the same color. Visualized audio samples are provided in §Appendix A.2 and audible samples are available on `demo page`.

| | quantization | bitrate | ASR-WER↓ | | UTMOS↑ | | Spkr. Sim.↑ | | ViSQOL↑ | |
|---|---|---|---|---|---|---|---|---|---|---|
| | | | dev | test | dev | test | dev | test | dev | test |
| Ground Truth | - | 256k | 3.2 | 3.4 | 3.87 | 3.77 | - | - | - | - |
| Encodec [a,f] | $VQ_{1024}$ | 500 | 41.7 | 41.0 | 1.24 | 1.24 | 0.25 | 0.25 | 2.96 | 2.88 |
| | $RVQ_{2\times1024}$ | 1k | 9.6 | 10.3 | 1.52 | 1.50 | 0.60 | 0.60 | 3.59 | 3.47 |
| | $RVQ_{8\times1024}$ | 4k | 3.8 | 4.0 | 2.94 | 2.82 | 0.89 | 0.89 | 4.25 | 4.21 |
| DAC [b,f] | $VQ_{1024}$ | 500 | 80.9 | 80.6 | 1.24 | 1.24 | 0.11 | 0.10 | 1.88 | 1.73 |
| | $RVQ_{2\times1024}$ | 1k | 22.8 | 24.2 | 1.28 | 1.28 | 0.33 | 0.31 | 2.95 | 2.70 |
| | $RVQ_{8\times1024}$ | 4k | 3.8 | 4.1 | 3.09 | 3.03 | 0.87 | 0.87 | 4.42 | 4.39 |
| $HuBERT_{BASE}$ [c,g] | $KM_{1024}$ | 500 | 9.8 | 10.2 | 2.36 | 2.37 | 0.14 | 0.15 | 2.14 | 2.15 |
| $HuBERT_{LARGE}$ [c,g] | $KM_{1024}$ | 500 | 7.6 | 7.9 | 2.66 | 2.65 | 0.14 | 0.15 | 2.00 | 2.03 |
| SpeechTokenizer [d,g] | $VQ_{1024}$ | 500 | 10.5 | 11.0 | 1.25 | 1.26 | 0.16 | 0.16 | 1.97 | 2.03 |
| | $RVQ_{2\times1024}$ | 1k | 8.5 | 9.1 | 2.20 | 2.08 | 0.34 | 0.34 | 3.04 | 3.03 |
| | $RVQ_{8\times1024}$ | 4k | 4.5 | 4.9 | 3.65 | 3.56 | 0.83 | 0.83 | 4.22 | 4.20 |
| UniWav [g] | $KM_{1024}$ | 500 | 7.2 | 7.1 | 3.64 | 3.59 | 0.25 | 0.26 | 2.14 | 2.09 |
| | $KM_{2\times1024}$ [e] | 1k | 5.6 | 5.6 | 3.79 | 3.72 | 0.49 | 0.50 | 2.96 | 2.92 |

[a]Défossez et al. (2022); [b]Kumar et al. (2024); [c] With unit-HiFiGAN Polyak et al. (2021); Nguyen et al. (2023); [d]Zhang et al. (2023b); [e] Residual quantization, see Eq.(11); [f] Trained on general audio; [g] Trained on LibriSpeech only;

tasks like speech recognition. To ablate the effectiveness of our pre-training method, we report baseline numbers without pre-training. Our result suggest that pre-training has more impact on the model's acoustic modeling ability (measured by SIM) and less on the language modeling ability (measured by ASR-WER).

## 3.4 SPEECH TOKENIZATION AND RESYNTHESIS

**Task introduction** In addition to purely discriminative and generative tasks, there are tasks that bridge the gap between the two, such as speech tokenization and resynthesis. With the recent advancements in large language models, there has been growing interest in converting speech into sequences of discrete tokens (Zhang et al., 2023b; Zhu et al., 2023; Huang et al., 2023; Bai et al., 2024), allowing speech to be processed similarly to text. These tokens must also be capable of being decoded back into audio signals, enabling token-based language models to generate speech.

**Prior works** We consider four prior works Encodec (Défossez et al., 2022), DAC (Kumar et al., 2024), unit-HiFiGAN based on HuBERT units (Hsu et al., 2021; Polyak et al., 2021; Nguyen et al., 2023), and SpeechTokenizer (Zhang et al., 2023b). Encodec and DAC are general neural audio codec models that are not designed specifically for speech tokenization. HuBERT units and Unit-HiFiGAN (Polyak et al., 2021) is a cascaded system for speech tokenization and resynthesis. Tokenization is done by running kmeans clustering on the representation extracted from a pre-selected layer of HuBERT. Resynthesis is done by training HiFiGAN (Kong et al., 2020) that takes speech tokens as input. For Unit-HiFiGAN, we train the model using the codebase released by the authors (Nguyen et al., 2023). In addition to $HuBERT_{BASE}$ used in the original work, we also report results using $HuBERT_{LARGE}$ that matches the size of UniWav encoder. To ensure a consistent setup, we used 1024 kmeans clusters and did *not* include speaker and emotion conditions to match other models. SpeechTokenizer is a neural audio codec model based on Encodec. In addition to audio reconstruction and adversarial training used in Encodec, SpeechTokenizer introduced an additional loss to distill representation from a pre-trained HuBERT into the first vector quantization layer. For SpeechTokenizer, we used the official model released by the authors for evaluation. Both methods required hand-selecting a layer from HuBERT for quantization or distillation, which is done by identifying the layer that matches the underlying phone label the most. Prior works therefore referred representations from the selected layer to as *semantic* representation. As a reference, both unit-HiFiGAN and SpeechTokenizer use layer 9 of $HuBERT_{BASE}$ to extract semantic representation, and we found layer 22 works best for $HuBERT_{LARGE}$. While unit-HifiGAN, SpeechTokenizer, and our model are trained on LibriSpeech, Encodec and DAC are trained on a mixture of speech, sound, and music.

Table 3: Recognition WER of different model sizes. Results are on dev-other with LM after 200k pre-training steps.

| encoder depth | decoder depth | | |
|---|---|---|---|
| | 0 | 12 | 24 |
| 12 | **7.6** | 8.1 | 8.2 |
| 24 | 5.8 | 5.5 | **5.2** |

Table 4: In-context TTS ASR-WER and speaker similarity of different model sizes after 200k pre-training steps. The default encoder/decoder depth is 24/12.

| decoder depth | encoder depth | | |
|---|---|---|---|
| | 0 | 12 | 24 |
| 12 | 3.4 / 0.45 | 2.5 / 0.60 | 2.5 / 0.63 |
| 24 | 2.5 / 0.61 | 2.5 / 0.63 | 2.5 / 0.63 |

**Fine-tuning UniWav**    UniWav can be naturally applied to the task by introducing a discrete bottleneck between the encoder and decoder. We follow the prior works to first identify the semantic layer to run kmeans. More information can be passed to the decoder by quantizing the remaining layers at a cost of a higher bitrate for tokenization. More precisely, we replace the encoder representations $z$ in Eq.(9) with

$$\bar{z} = W_i \underbrace{\operatorname*{argmin}_{q \in Q_i} \left\| z^{(i)} - q \right\|_2}_{\text{semantic quantization}} + \underbrace{\operatorname*{argmin}_{q \in Q_{\text{residual}}} \left\| \sum_{j \neq i} W_j z^{(j)} - q \right\|_2}_{\text{residual quantization (optional)}}, \tag{11}$$

where $i = 10$ is the index of layer used for extracting semantic representation, $Q_i$ and $Q_{\text{residual}}$ are the sets of centroids obtained through k-means clustering. We fine-tune the decoder with the quantized encoder representation on 8 A100 GPUs for 150k steps with a 1e-4 learning rate. For decoding, we use a step size of 0.25 and CFG $\alpha = 1.0$, resulting in 8 NFEs.

**Data & Metrics**    All models considered are trained on LibriSpeech and tested on the `dev` and `test` subsets including both `clean` and `other` splits. For audio intelligibility, we report ASR-WER (see §3.3) on the resynthesized speech. For the overall audio quality, we use UTMOS (Saeki et al., 2022) following the discrete vocoder training track of the Interspeech 2024 Challenge on Speech Processing Using Discrete Units (Chang et al., 2024). UTMOS is a reference-free, machine perception score on audio naturalness on a five-point scale that aligns well with human preference. We also measure and speaker similarity (see §3.3) between the original audio and the resynthesized version. For audio reconstruction quality, we measure ViSQOL (Chinen et al., 2020) following Encodec and DAC.

**Results**    Results are presented in Table 2. At 500 bps, UniWav leads in all metrics by a margin, outperforming all other methods. The lowest ASR-WER reflected that UniWav carried the most semantic information under the same bitrate. Outstanding UTMOS suggested UniWav is able to generate realistic audio with limited information provided, which emerges from the generative pre-trained decoder. We note that speaker information is still dropped by UniWav, much like all other low-bitrate methods, due to the choice of semantic representation. At 1k bitrate, UniWav maintained a clear advantage over SpeechTokenizer, with a significantly lower word error rate (5.6% v.s 9.1%) and higher UTMOS (3.72 v.s. 2.08). In fact, 1kbps UniWav is comparable to 4kbps SpeechTokenizer in terms of ASR-WER and UTMOS. UniWav only falls short on speaker similarity, which requires input speaker information that cannot be filled in by the decoder. Nevertheless, the similarity score is still significantly higher (0.499 v.s. 0.340) than SpeechTokenizer at the same 1kbps.

Results on speech tokenization and resynthesis provided strong evidence for the usefulness of jointly learning speech representation and generation. UniWav is able to perform low-bitrate speech encoding while decoding audio of substantially better quality than the state-of-the-art SpeechTokenizer.

### 3.5    ANALYSIS

**Scaling Encoder and Decoder**    Since the encoder and the decoder are trained jointly in UniWav, it is interesting to investigate how the scale of the encoder and the decoder changes the overall behavior. In Table 3 and Table 4, we report speech recognition and synthesis results when varying the size of the encoder and the decoder. Zero depth indicates the corresponding training objective was ignored. For speech recognition with a shallow encoder, we found introducing the decoder degrades WER regardless of the size. Interestingly, an opposite trend is observed when the encoder size is doubled, i.e., the encoder benefits from the decoder and the generative objective. For speech generation, we found that encoder depth was critical when working with a shallow decoder, and the gain became less

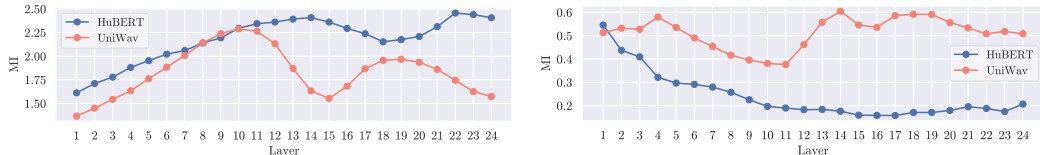

Figure 2: Mutual information between quantized representation and phone/speaker (left/right) at different layers. Results are computed on the dev set of LibriSpeech. For quantization, 1024 clusters are used for k-means.

significant with a deeper decoder. More importantly, the size of the decoder became less important as the encoder depth increased.

To summarize, we highlight two important observations regarding unified pre-training:

- Discriminative representation learning benefits from generative pre-training only when the encoder has enough capacity.
- Representation encoder plays a more important role in unified pre-training, offering a better trade-off between compute and performance.

**Mutual Information Between Representation and Labels**  For speech representation learning, layer-wise analysis has been used to examine what is encoded by the model and its dynamics at different layers (Pasad et al., 2023). In this study, we introduce a discrete bottleneck to each layer of UniWav's encoder, using the method described in §3.4 and Eq.(11). For each layer, we approximate the mutual information (MI) between quantized representations and the underlying speech label — either phone or speaker — by computing the empirical distribution $P(\text{unit})$, $P(\text{label})$, and $P(\text{unit}, \text{label})$, and visualize them in Figure 2. The results provide explanations for UniWav's behavior in different applications: (1) In the latter layers, UniWav achieved lower MI in relation to phones compared to HuBERT, potentially making fine-tuning for speech recognition more difficult; (2) For most of the layers, UniWav retained a higher MI in relation to speaker label, which explained the strong performance on audio quality and speaker similarity metrics in speech generation and resynthesis.

Through the analysis, we conclude some observations and future directions: (1) Due to the existence of the generative decoder, UniWav captured information beyond language from speech. Such property might overturn the belief that self-supervised speech representations are sub-optimal  (Wang et al., 2023) for generation tasks. Our speech tokenization experiment is a good evidence. (2) Compared to the pure self-supervised learning algorithm, UniWav learns a more entangled representation. Future works in unified pre-training can potentially benefit from representation disentanglement methods (Qian et al., 2022; Chang et al., 2023) to further improve controllability and downstream performance.

## 4  RELATED WORKS

The success of self-supervised representation learning and generative models has emerged as a promising direction for enhancing model generalization, leading to general-purpose speech and audio-intelligent systems. Our work contributes to this growing body of research by rethinking the potential of jointly modeling both speech understanding and generation, aiming for a unified framework that can seamlessly integrate these tasks. We would like to acknowledge several previous efforts in this space, particularly the intersection between speech representation learning and generative modeling.

**Speech Representation Learning**  In recent years, the research community has witnessed significant advancements in self-supervised learning techniques across speech and other domains, with notable developments ranging from vector-quantized methods (Van Den Oord et al., 2017; Dhariwal et al., 2020), contrastive pre-training methods (Schneider et al., 2019; Baevski et al., 2020b), to more recent random-projection quantizer methods (Chiu et al., 2022; Yang et al., 2024) or self-labeling methods (Hsu et al., 2021; Chen et al., 2022a; Baevski et al., 2022; Liu et al., 2024). These self-supervised representation learning methods have a wide impact in speech processing, but their focus remained primarily on discriminative tasks. Applying these techniques to generative tasks is more

challenging (Tsai et al., 2022), often relying on a pipelined generative model (Polyak et al., 2021; Zhang et al., 2023b). While UniWav's encoder follows this line of work, our experiments and analysis suggested that it is significantly different from prior work, due to the generative components involved in pre-training. UniWav can be viewed as a more general representation framework aiming to support a more diverse collection of applications.

Besides speech-only models, it is also worth noting that text-injection-based pre-training could improve speech and text representation on cross-modality matching (Chung et al., 2019; Chen et al., 2021; Ao et al., 2021) and feature disentanglement (Chen et al., 2022b). Unified pre-training can potentially be further improved by cross-modality alignment in future studies.

**Generative Models in Speech**    Similar to representation learning, generative models have also made significant strides in the field of speech. Methods like adversarial training (Kong et al., 2020; Lee et al., 2022), normalizing flows (Prenger et al., 2019; Valle et al., 2020), auto-regressive models (Wang et al., 2023; 2024), and diffusion-style framework (Koizumi et al., 2022; Le et al., 2023; Kim et al., 2024) have all pushed the edge of speech generation. In this line of work, generative pre-training with Flow Matching (SpeechFlow; Liu et al. 2023) is the most relevant work. Our UniWav decoder is built upon SpeechFlow's design and follows a similar fine-tuning paradigm. Our decoder can be viewed as a representation-conditioned variant of SpeechFlow. Nevertheless, we note that the jointly optimized representation encoder and generative decoder are substantially different from training generative models on fixed embeddings, and our contribution in this direction is novel.

**Speech Foundation Models and Large Langauge Models (LLM).** In alignment with the recent trend in audio-enabled LLMs (Gong et al., 2023; Tang et al., 2023; Zhang et al., 2023a; Kong et al., 2024; Huang et al., 2024), representation learning and generative models have gained further popularity. For such models, we often desire them to possess the capability of both perceiving (discriminative tasks) and producing (generative tasks) audio content, e.g., GPT-4o (OpenAI, 2024), and Moshi (Défossez et al., 2024). Furthermore, modern LLMs are typically trained in an autoregressive manner, and thus require the model's input and output to be in a *unified* representation space. However, previous discriminative representations, such as HuBERT (Hsu et al., 2021), are not easily adaptable to generative tasks. Conversely, representations with more detailed acoustic features, such as EnCodec codes (Défossez et al., 2022), exhibit limitations in discriminative performance and degraded performance in phonetic classification tasks (Yang et al., 2023). As a result, prior models either do not support sound generation (Gong et al., 2023; Tang et al., 2023; Kong et al., 2024), rely on additional components like a vocoder (Zhang et al., 2023a) or text-to-speech model (Huang et al., 2024), or has a higher word error rate than models trained with discriminative representations (Défossez et al., 2024). This highlights the urgent need for a speech representation that excels in both discriminative and generative tasks. UniWav addresses this gap by introducing a unified representation, which is essential for building future foundation models that integrate both understanding and generation capabilities.

## 5    CONCLUSION

In this work, we introduced UniWav, a unified framework for speech representation learning and generation. By integrating both discriminative and generative capabilities, UniWav demonstrates that a single model can effectively bridge the gap between tasks with competing requirements. Through experiments across speech recognition, synthesis, and tokenization tasks, we show that UniWav achieves competitive performance compared to task-specific models, reducing the need for distinct pre-trained models and simplifying the speech processing pipeline. The compactness of the learned representation and the ability to generate high-quality audio underscore the potential of unified foundational models for future Speech and Audio intelligent systems.

**Limitations**    The most significant limitation we noticed is the performance drop on recognition tasks compared to self-supervised learning models at similar scales. To the extent of our experimentation thus far, the cost of enabling generation appears to make this result inevitable. In computer vision, researchers have made similar observations on the small gap between a unified model and a single-function state-of-the-art representation model (Li et al., 2023). Whether the gap can be closed or not remains an open research problem. We also note that this work focused exclusively on English Speech, from audiobooks in particular. Applying UniWav to more generic speech, more languages, and different audio domains (such as sound and music) are left as important future works.

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

# A  APPENDIX

## A.1  ADDITIONAL EXPERIMENTS

Table 5: Comparison of the choice of surface feature. Models are pre-trained for 200k steps. Sim-r refers to the similarity when comparing generated speech with the resynthesized prompt, whereas Sim-o uses original prompt audio as the reference.

| Feature | Recognition | Speech Synthesis | | |
|---|---|---|---|---|
| | WER | ASR-WER | Sim-r | Sim-o |
| Mel Spectrogram | | | | |
| Decoding ground truth | - | 2.6 | 0.638 | 0.666 |
| UniWav | 5.6 | 2.5 | 0.629 | 0.620 |
| EnCodec latent | | | | |
| Decoding ground truth | - | 2.5 | 0.627 | 0.666 |
| UniWav | 5.5 | 2.5 | 0.631 | 0.614 |

**Choice of Surface Feature**   Different speech foundation models have been relying on different input representations of audio, ranging from time-domain signal (Baevski et al., 2020a;b; Hsu et al., 2021), mel spectrogram (Chung et al., 2021; Le et al., 2023; Liu et al., 2023), to neural auto-encoder features (Wang et al., 2023; Vyas et al., 2023; Evans et al., 2024). In this work, we choose Encodec (Défossez et al., 2022) latent since it can be easily decoded, making it ideal for audio generation. Besides Encodec latent, we also train UniWav on mel spectrogram to ablate the choice of audio feature in Table 5. 80-band mel spectrogram is extracted at a 50Hz frame rate (to match Encodec latent) with the hop size of 320 and window size of 1280. For vocoding, we adopted the BigVGAN-v2 vocoder (Lee et al., 2022) by modifying its configuration to accept an 80-band mel spectrogram at a 50Hz frame rate and generate a 16kHz waveform. Following the EnCodec architecture, we use four residual blocks with upsampling ratios of [8, 5, 4, 2] resulting in a 122M parameter model. We adjust the `bins_per_octave` of the MS-SB-CQT Discriminator (Gu et al., 2024) to [6, 10, 14] to ensure all values remain below the Nyquist frequency for 16kHz audio. We use a learning rate of $5 \times 10^{-5}$ and a batch size of 32, using 2-second chunks per sample.

On both the recognition and synthesis tasks, we also observe similar results regardless of the model input. By decoding the mel spectrogram and Encodec latent extracted from ground truth, we also found that both systems shared similar baseline performance on the quality of generation. We conclude that the proposed pre-training framework is general and robust to the choice of surface feature.

Table 6: Speech representation evaluation results on SUPERB (Yang et al., 2021). Results of prior works are taken from the official leaderboard.

|  | Pre-training data (hr) | Phoneme Recognition Phone Error Rate | Speaker Identification Accuracy |
|---|---|---|---|
| HuBERT (Hsu et al., 2021) | 60k | 3.5 | 90.3 |
| WavLM (Chen et al., 2022a) | 94k | 3.1 | 95.5 |
| UniWav | 60k | 3.7 | 93.8 |

**Speech Representation Evaluation** To study the representation extracted by UniWav encoder, we conduct experiments on phone recognition and speaker identification using the existing benchmarking toolkit SUPERB (Yang et al., 2021), results are provided in Table 6. We use the default configuration provided by the SUPERB toolkit [2], sweeping only the learning rate for both tasks. In these tasks, representations are kept frozen to train downstream models, simulating the use case where a pre-trainined representation model is treated as a feature extractor. Results on SUPERB show that representation extracted from UniWav performed slight worse than that of from HuBERT on phoneme recognition, but better on speaker identification. WavLM, the leading model on SUPERB, outperforms both models with more training data and intensive hyper-parameter search on downstream tasks (Chen et al., 2022a). In short, the conclusion here is consistent with our experiments on fine-tuning for speech recognition (§3.2) and layer-wise analysis of speech representation (§3.5).

Table 7: Subjective test on English zero-shot speaker adaptation TTS on filtered LS test-clean. Averaged rating along with 95% confidence interval are reported for Mean Opinion Score (MOS).

| Method | subjective MOS |
|---|---|
| Ground truth | 3.82±0.11 |
| SpeechFlow (Liu et al., 2023) | 3.59±0.11 |
| UniWav | 3.55±0.11 |

**Subjective Evaluation on Text-to-speech** Following the evaluation setup of Liu et al. (2023), we conducted subjective test on text-to-speech results using mean opinion score (MOS) (Ribeiro et al., 2011). We randomly selected 50 sentences from the test-clean subset of LibriSpeech for the test and collected 10 ratings for each sentence. Each annotator is asked to rate 30 samples, evenly distributed across ground truth, SpeechFlow (Liu et al., 2023), and UniWav. The ratings are based on the clarity of speech, sound quality, and naturalness, ranging from 1 to 5. Results are collected through Amazon Mechanical Turk (AMT) and the listed in Table 7. We found UniWav recieving similar rating to that of SpeechFlow, while both methods fall short of the ground truth. The MOS score supported our conclusion from objective tests (see Table 1) that UniWav matches the performance of generative-only state-of-the-art pre-training method on text-to-speech.

---

[2]https://github.com/s3prl/s3prl/tree/main

## A.2 VISUALIZED SAMPLES OF SPEECH TOKENIZATION AND RESYNTHESIS

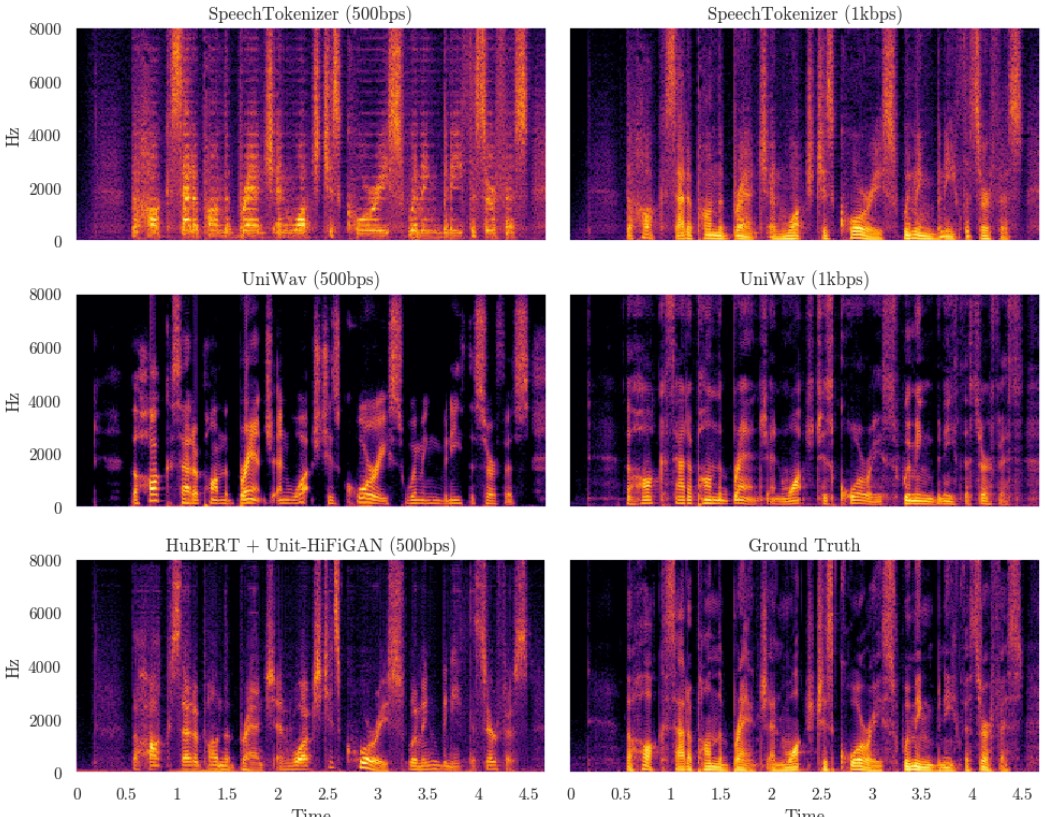

Figure 3: Utterance `7176-88083-0014` from `test-clean`

