# OpenReview forum: "UniWav: Towards Unified Pre-training for Speech Representation Learning and Generation"
_ICLR.cc/2025/Conference — ICLR 2025 Poster_

### Official Review · Reviewer_v3oX · 2024-10-18

**Soundness:** 3
**Presentation:** 3
**Contribution:** 2
**Rating:** 6
**Confidence:** 5

**Summary:**

This paper proposes a framework named UniWav. It is an encoder-decoder framework aimed at unifying pre-training representation learning and generative tasks. The authors claimed that this is the first such framework and achieves comparable performance to different existing foundation models, each trained on a specific task, on speech recognition, text-to-speech, and speech tokenization tasks.

**Strengths:**

The main contribution of this work is the combination of DinoSR and a flow matching model to train a unified representation model. While not considered significant, this special setup is somewhat novel.

The paper contains sufficient experimental results to support their claims although some setups are questionable (in detail below).

The paper clearly indicated the limitations of the work.

**Weaknesses:**

The claim their work is "the first unified pre-training framework for speech representation learning and generation" is arguable. For example, the paper "Speechgptgen: Scaling chain-of-information speech generation." introduced an approach to train a discrete representation that works for both understanding and generation. The paper "Moshi: a speech-text foundation model for real-time dialogue" further improved the approach.

The presentation needs to be improved.
1. the DinoSR part in Figure 1 is confusing since the softmax layer is not clearly shown since the encoder, according to the text description, does not include that layer. In contrast, the figure in the original DinoSR paper is very clear.
2. The majority of Section 2.1 is describing DinoSR. However, notation is not clearly explained sometimes. for example, what is "k" in s^k_v right below eq. 3? it's unclear why Euclidean distance (instead of COS distance) is used given that Cos distance is usually more preferred in higher-dim spaces.
3. notation "z" is overloaded in eq 9.
4. the footnote under Table 1 is not clear. alignment is only used for generation tasks?
5. eq 11 is wrong? argmin returns an index instead of a representation.

When researchers train a representation model they usually keep the encoder fixed when using them in downstream tasks such as ASR. however, in this work the encoder is finetuned. This causes the claims weaker. Similarlly if flow matching is used in the generation part the quality of the generated speech will of course become better. However, this gain comes from the flow matching model instead of the way the representation is learned. Some clarification here is needed.

**Questions:**

why "For speech recognition with a shallow encoder, we found introducing the decoder degrades WER regardless of the size. Interestingly, an opposite trend is observed when the encoder size is doubled"? This is something requires more analysis and explanation.

In section 3.1, "We extract the prequantized latent feature from a 16kHz EnCodec (D´efossez et al., 2022) encoder, pre-trained on the same dataset, to serve as the input for our model." This is contradictory to the claim in early sections that the input to your model is mel spectrum?  In addition, this makes the comparison with other approaches unfair since you used another model as the pre-processing module (i.e., total model size is actually increased significantly).

---

> ### Author Response · Authors · 2024-11-24
>
> > The claim their work is "the first unified pre-training framework for speech representation learning and generation" is arguable. For example, the paper "Speechgptgen: Scaling chain-of-information speech generation." introduced an approach to train a discrete representation that works for both understanding and generation. The paper "Moshi: a speech-text foundation model for real-time dialogue" further improved the approach.
>
> We thank the reviewer for raising the question on our claim. We would like to clarify that both  SpeechGPTGEN and Moshi are multi-modal (speech-text) pre-trained model, whereas UniWav focused on single-modal pre-training. Multi-modal models like SpeechGPTGEN and Moshi rely on single-modal pre-trained backbones (e.g., SpeechTokenizer in SpeechGPTGEN and Mimi codec in Moshi both relied on speech self-supervised model), and our work can be viewed as the upstream pre-training to serve for multi-modal models.
>
> ---
>
> > the DinoSR part in Figure 1 is confusing since the softmax layer is not clearly shown since the encoder, according to the text description, does not include that layer. In contrast, the figure in the original DinoSR paper is very clear.
>
> Figure 1 did not include details like the softmax layer for simplicity, we are sorry if it is causing confusion. The prediction head and softmax are listed explicitly in line 126, which also appears in the overall objective of encoder (Eq.4).
>
> > The majority of Section 2.1 is describing DinoSR. However, notation is not clearly explained sometimes. for example, what is "k" in s^k_v right below eq. 3? it's unclear why Euclidean distance (instead of COS distance) is used given that Cos distance is usually more preferred in higher-dim spaces.
>
> The “k” was a legacy typo from earlier version of the draft, we have removed it. We thank the reviewer for pointing it out. For measuring distance in high-dim space, we believe Euclidean distance is equally common for clustering (e.g., kmeans), assuming there is no special normalization before quantization.
>
> > the footnote under Table 1 is not clear. alignment is only used for generation tasks?
>
> We thank the reviewer for raising the question, we have add more words to clarify it.
>
> > eq 11 is wrong? argmin returns an index instead of a representation.
>
> Although argmin in python returns index, in math notation Argmin is formally defined as the function that returns the element of a set which provides the minimum value of some criterion (https://en.wikipedia.org/wiki/Arg_max).
> In short, argmin in our case returns the centroid, which is a vector, that is closest to the input embedding.
>
> ---
>
> > When researchers train a representation model they usually keep the encoder fixed when using them in downstream tasks such as ASR. however, in this work the encoder is finetuned. This causes the claims weaker. Similarlly if flow matching is used in the generation part the quality of the generated speech will of course become better. However, this gain comes from the flow matching model instead of the way the representation is learned. Some clarification here is needed.
>
> We thank the reviewer for the helpful comments, **we have added experiments where speech representation is kept frozen (i.e., the encoder is not finetuned) using the existing speech representation benchmark to Section A1**.
>
> ---
>
> > In section 3.1, "We extract the prequantized latent feature from a 16kHz EnCodec (D´efossez et al., 2022) encoder, pre-trained on the same dataset, to serve as the input for our model." This is contradictory to the claim in early sections that the input to your model is mel spectrum? In addition, this makes the comparison with other approaches unfair since you used another model as the pre-processing module (i.e., total model size is actually increased significantly).
>
> In line 108, we use mel spectrogram as an example for surface feature. In practice, our method works with either Encodec latent or mel spectrogram and encodec latent. As shown in Appedix A.1, the choice of input feature (spectrogram or Encodec feature) have little to no impact on the behavior of the model on ASR/TTS. We are sorry for the confusion, and have edited line 108 accordingly.

---

### Official Review · Reviewer_hgj6 · 2024-10-31

**Soundness:** 3
**Presentation:** 3
**Contribution:** 2
**Rating:** 6
**Confidence:** 4

**Summary:**

This paper introduces UniWav, a new pre-training framework focused on both speech representation learning for discriminative task and for generation task. Using an encoder-decoder architecture, UniWav learns speech representations and generates speech together, letting it perform well in both discriminative tasks and generative tasks. UniWav aims to unify the approach, simplifying the speech processing pipeline and reducing need for many specialized pre-trained models. Experiments in speech recognition, text-to-speech synthesis, and speech tokenization show UniWav's effectiveness, achieving results competitive with other top models for specific tasks.

**Strengths:**

The paper is well motivated to jointly pre-train for discriminative and generative tasks. The paper shows SOTA, if not comparable, results on the generative tasks and lag a little on the discriminative task.
They show very good results when tokenizing speech using UniWav, compared to other works in the literature.
The paper is well written.

**Weaknesses:**

The authors propose UniWav, a pre-training method for jointly learning representations for both discriminative and generative tasks in speech processing. This method uses an encoder-decoder architecture to learn effective representations as part of the training process.

For the discriminative task of speech recognition, the authors fine-tune the encoder on 100-hour and 960-hour splits, showing competitive results compared to other methods. However, they do not provide results for low-resource scenarios (e.g., 10 hours or 1 hour of data) or results using a frozen encoder (SUPERB benchmark). These two setups are essential to support their claim that the learned encoder representations are effective for speech recognition tasks. Because with large finetuning data, effect of pre-training is less.

Figure 2 shows that the encoder’s full capacity is not optimized for learning features specific to speech recognition. UniWav achieves the highest mutual information (MI) on layer 10 out of 24, which then gradually decreases. This pattern suggests that the model divides its capacity between the discriminative and generative tasks, similar to observations in WavLM and prior works. This could explain UniWav’s strong performance in speech generation tasks, as a significant portion of the model’s capacity is allocated to optimize for generation. This trend is consistent across the paper’s reported results.

Regarding speech tokenization, it’s unclear if the comparison is entirely fair. For speech tokenization, the input to the SpeechTokenizer is raw audio, while for UniWav, the input is Encodec encoder features, which are further transformed by UniWav’s 10-layer encoder. It is as if UniWav is heavily overparameterized and most of the parameters are used for generation task.

**Questions:**

On line 108, the authors claim that the model uses mel spectrograms as input, but on lines 239–240, they mention using Encodec features instead.

---

> ### Author Response · Authors · 2024-11-24
>
> > For the discriminative task of speech recognition, the authors fine-tune the encoder on 100-hour and 960-hour splits, showing competitive results compared to other methods. However, they do not provide results for low-resource scenarios (e.g., 10 hours or 1 hour of data) or results using a frozen encoder (SUPERB benchmark). These two setups are essential to support their claim that the learned encoder representations are effective for speech recognition tasks. Because with large finetuning data, effect of pre-training is less.
>
> We thank the reviewer for the helpful comments, **we have added experiment using SUPERB to Section A1**. Due to time constraint, we selected phone recognition and speaker identification for evaluating speech representation. In short, results are consistent with our findings in speech recognition and layer-wise analysis.
>
> ---
>
> > Regarding speech tokenization, it’s unclear if the comparison is entirely fair. For speech tokenization, the input to the SpeechTokenizer is raw audio, while for UniWav, the input is Encodec encoder features, which are further transformed by UniWav’s 10-layer encoder. It is as if UniWav is heavily overparameterized and most of the parameters are used for generation task.
>
> We thank the reviewer for raising the concern and would like to clarify the use of parameters. While UniWav builds on top of a pre-trained Encodec model, SpeechTokenizer also distills from a pre-trained WavLM. So from a high-level perspective, both models required a CNN-based encoder plus a deep transformer to effectively learn discrete tokens.
>
> It is also worth mentioning that results in Appedix A.1 have suggested that the choice of input feature (spectrogram or Encodec feature) have little to no impact on the behavior of the model on ASR/TTS. Based on the results, the # of parameter used in total can likely be further reduced by switching to spectrogram.
>
> ---
>
> > On line 108, the authors claim that the model uses mel spectrograms as input, but on lines 239–240, they mention using Encodec features instead.
>
> As shown in Appedix A.1, the choice of input feature (spectrogram or Encodec feature) have little to no impact on the behavior of the model on ASR/TTS. We are sorry for the confusion, and have edited line 108 accordingly.
>
> ---
>
> We hope the responses and updates on the paper have addressed the concerns raised by the reviewer.

---

> > ### Comment · Reviewer_hgj6 · 2024-12-02
> >
> > Thank you for the comments. I still share the same concerns raise by the reviewer g67C. In terms of performance, I do not see the speech community using this framework.  The encoder part of the model still divides its capacity while encoding different information types content and other, similar to the WavLM. I will keep my score.

---

### Official Review · Reviewer_me7k · 2024-11-02

**Soundness:** 3
**Presentation:** 4
**Contribution:** 3
**Rating:** 8
**Confidence:** 3

**Summary:**

This work proposes a self-supervised speech representation learning objective which combines 1) masked prediction with online clustering from an EMA teacher model (DinoSR) to train a transformer-encoder network and 2) reconstructing noise-inserted input data based on the encoder representations with Flow Matching to train a transformer-decoder network. The aim of this method is to unify the creation of foundation models for discriminative tasks (such as ASR) and generative tasks (such as TTS).

The method is evaluated by pre-training on 60k hours of LibriLight, and fine-tuning for speech recognition, speech synthesis, and speech tokenization and resynthesis, on Librispeech.

For speech recognition, they show limited degradation of performance compared to SSL methods like HuBERT, WavLM, and data2vec.
For speech synthesis, they show performance matching contemporary models like VoiceBox and SpeechFlow.
For speech tokenization and reconstruction, performance exceeds SpeechTokenizer and HuBERT+HifiGAN.

The method is also ablated on the encoder and decoder depth, which shows that a 12-layer encoder does not benefit from adding the decoder objective, while a 24-layer encoder does benefit. Moreover, it is shown through a mutual-information analysis that their encoder has different characteristics on how speaker and speech information is processed compared to HuBERT.

**Strengths:**

### Originality

The method proposes a new speech SSL method, combining existing methods DinoSR (encoder-only) and SpeechFlow (encoder-decoder) . This method has strong performance on generative and discriminative tasks compared to foundation models which are generative-only or discriminative-only.

### Quality

The method is evaluated on multiple speech technology tasks, and a small ablation study is performed for further insights.

### Clarity

The paper is well-written, easy to follow, and appropriately places itself into existing literature.

### Significance

This work will definitely spark future work in the speech community.

**Weaknesses:**

### domain
This works only evaluates on Librispeech, i.e., the audio book domain, which has very homogenetic speaker conditions. The observation that ASR performance closely matches HuBERT, WavLM, etc, does not take into account the robustness these models have to other domains. The authors could discuss this limitation or add experiments with the SUPERB benchmark.

### modelling
As the proposed method is an extension of DinoSR, it would be nice to see UniWav w/o decoder, as in Table 3 for 200k steps, in Table 1 with 600k steps of pre-training. Moreover, Speechflow also uses an encoder-decoder model. This encoder could in theory be fine-tuned for ASR. The original work did not do this. Can the authors comment on expected results when the SpeechFlow encoder would be fine-tuned for ASR, and how this would compare to UniWav? Could this be discussed in a future work section?

### speaker recognition

One of the central claims is that UniWav bridges the inherent orthogonality between speech recognition, which normalizes over speaker and environment information, and speech synthesis, which requires speaker and environment information. This is touched upon slightly by the mutual information analysis, where UniWav is seen to lose speaker information through the encoder layer. It would seem to me that UniWav will not be better for the speaker recognition task than, e.g., WavLM, based on the speculation that most environment and speaker information is stored in the decoder. I think evaluating UniWav on the (SUPERB) speaker recognition and speaker verification task would strengthen the claim significantly. I think for now, this limitation should be made more explicit in the limitation section, or if possible, the authors could perform additional experiments on SUPERB.

### Minor comments

1. line 215: Encodec has not been introduced yet, so cite it here. I find this paragraph confusing due to missing context on how the network uses Encodec as input features (line 239) instead of Mel spectrograms as suggested in line 108.
2. line 236: \proposed~is
3. line 242: in~\ref
4. line 358: kmeans instead of k-means
5. line 370: we follow...run k-means, use e.g., by first identifying .. on which to run

**Questions:**

1. Based on Figure 2, have the authors considered only conditioning the decoder on the first, e.g., 12 layers of the encoder?
2. Can the authors clarify how speaker similarity is computed? Is it the average cosine similarity between the WavLM speaker embedding of pairs of ground-truth and synthesized utterances? If so, I think it would make sense to share the standard deviation as well.
3. Can the authors comment on their batch size and how they are sampled (e.g., like wav2vec 2.0)? This would help with reproducibility, and figuring out how much data is seen throughout pre-training, and where UniWav lies in Figure 2 of the DinoSR paper (trade-off between performance and data efficiency)
4. Line 233, "sinusoidal pos. enc. is appended to the input." Is this appended to the feature dimension or the time dimension? Isn't it normally the case that positional embeddings are summed with the input features? Can the authors comments on this design decision?

---

> ### Author Response · Authors · 2024-11-24
>
> > domain
>
> > This works only evaluates on Librispeech, i.e., the audio book domain, which has very homogenetic speaker conditions. The observation that ASR performance closely matches HuBERT, WavLM, etc, does not take into account the robustness these models have to other domains. The authors could discuss this limitation or add experiments with the SUPERB benchmark.
>
> We thank the reviewer for the helpful comments, we have added more discussion on the limitation in Section 5. **We have also added experiment using SUPERB to Section A1**. Due to time constraints, we selected phone recognition and speaker identification for
> evaluating speech representation. In short, the results are consistent with our findings in speech recognition and layer-wise analysis.
>
> ---
>
> > modelling
>
> > As the proposed method is an extension of DinoSR, it would be nice to see UniWav w/o decoder, as in Table 3 for 200k steps, in Table 1 with 600k steps of pre-training. Moreover, Speechflow also uses an encoder-decoder model. This encoder could in theory be fine-tuned for ASR. The original work did not do this. Can the authors comment on expected results when the SpeechFlow encoder would be fine-tuned for ASR, and how this would compare to UniWav? Could this be discussed in a future work section?
>
> For UniWav without decoder, we did not train it for full 600k updates, but instead trained more different combinations of encoder/decoder sizes to accommodate our compute budget, we hope the review can understand our rationale. We would also like to note that UniWav without decoder is still (slightly) different from DinoSR; the former is trained on Encodec feature and the latter relies on deep convolution that is trained from scratch jointly with the representation model.
>
> For fine-tuning SpeechFlow as ASR, we would first like to note that fine-tuning SpeechFlow as ASR can be less intuitive, for example: the input of the model, noise-infused speech and the time embedding specifically, is conditioned on the flow matching time step t. For discriminative tasks, it is unclear what is the meaning of sampling t.
>
> Nevertheless, we tried fine-tuning as the reviewer suggested. This is done by fixing t at zero. Using 100hr to fine-tune the model (as in Table 1), we observed that the word error rate failed to reach below 40%. Although there might be a better way (e.g., fix to a different t), our preliminary result suggested that generative-only pre-training is not useful for speech
>
> ---
>
> > run superb speaker recognition
>
> (done, see first section of response)
>
> ---
>
> > Questions
>
> > Based on Figure 2, have the authors considered only conditioning the decoder on the first, e.g., 12 layers of the encoder?
>
> We haven’t tried it, but this idea is interesting and worth exploring in the future.
>
> > Can the authors clarify how speaker similarity is computed? Is it the average cosine similarity between the WavLM speaker embedding of pairs of ground-truth and synthesized utterances? If so, I think it would make sense to share the standard deviation as well.
>
> Speaker similarity is computed through the cosine distance between the WavLM speaker embedding of ground truth and synthesized utterances. The variance of this score should be stable across different methods in general, we, therefore, omit it to save space in the table following the convention [A,B,C]. For your reference, the std of the score is 0.119/0.118 for SpeechFlow/Uniwav respectively.
>
>
> [A] https://arxiv.org/abs/2306.15687
>
> [B] https://arxiv.org/abs/2301.02111
>
> [C] https://arxiv.org/abs/2310.16338
>
> > Can the authors comment on their batch size and how they are sampled (e.g., like wav2vec 2.0)? This would help with reproducibility, and figuring out how much data is seen throughout pre-training, and where UniWav lies in Figure 2 of the DinoSR paper (trade-off between performance and data efficiency)
>
> We are sorry for missing the detail, the batch size is 312.5 seconds (5M frames at 16kHz) per GPU, where audio files are randomly sampled and randomly cropped into chunks no longer than 20 seconds. We have updated Section 3.1 to include the detail.
>
> > Line 233, "sinusoidal pos. enc. is appended to the input." Is this appended to the feature dimension or the time dimension? Isn't it normally the case that positional embeddings are summed with the input features? Can the authors comments on this design decision?
>
> Sinusoidal embedding is used to encode the time in ODE function (i.e., t in Eq.5), different from the common positional embedding used in transformers. It is treated as a single vector that is prepended to the input sequence of decoder, following the design of Voicebox and SpeechFlow. We are sorry for the confusion, we have make this more clear in Section 3.1.
>
> ---
>
> We would like to express our appreciation for the careful review and helpful comments, we have edited the submission to address issues and concerns raised in the review accordingly.

---

> > ### Comment · Reviewer_me7k · 2024-11-26
> >
> > I want to thank the authors for their clear reply and updating the work in such a short time-span. I've read the new appendix A.1 and the (reply to) other reviews. I think that this SSL method which can be fine-tuned for discriminate and generative tasks is interesting enough to warrant my score.

---

### Official Review · Reviewer_8mpX · 2024-11-03

**Soundness:** 2
**Presentation:** 3
**Contribution:** 2
**Rating:** 5
**Confidence:** 4

**Summary:**

The paper proposes UniWav, a unified pre-training framework for speech representation learning and generation. Traditionally, pre-training models for speech have been specialized either for discriminative tasks, like speech recognition, or for generative tasks, such as text-to-speech. UniWav aims to bridge this gap by integrating both functions into a single encoder-decoder model. The encoder is responsible for learning robust speech representations, while the decoder generates speech through Flow Matching.

**Strengths:**

1. The paper conducts experiments across multiple tasks like speech recognition, text-to-speech, and tokenization. It includes analyses, such as ablation studies and mutual information metrics.
2. The paper is well-organized and clearly presents technical details, such as the encoder-decoder structure and the Flow Matching method, making it easy to follow. Visual aids and concise explanations further contribute to the clarity of the complex concepts.

**Weaknesses:**

1. Firstly, I have some doubts regarding the motivation and novelty validation in this paper. The introduction states that ideally, speech representation learning and generation can mutually enhance each other and that this approach can reduce the overhead and cost of pre-training. However, I did not find evidence for these conclusions in the experimental section. Specifically, it seems that the paper does not address whether pre-training on just one task (either speech representation learning or generation) would yield better performance on downstream tasks compared to simultaneous pre-training on both tasks. Additionally, the model's performance on downstream tasks does not appear to be very strong; for instance, its performance on the speech recognition task is worse than that of other baselines. So the experiments can not show that the overhead and cost can be reduced. Because there is neither proof that pre-training on both tasks together is better, nor evidence that there are significantly better results on a single downstream task. If these advantages cannot be clearly demonstrated, then why combine the two tasks? Merely putting speech representation learning and speech generation together without thoroughly explaining the rationale and benefits of this approach significantly limits the contributions of the paper. Overall, the results seem to contradict the stated motivation and novelty, or these points have not been well validated.

2. Secondly, the selection of baselines in the paper is quite limited, especially for the speech generation and tokenization tasks. Speech generation could be compared with Natural Speech 3 [1], while the speech tokenization task could benefit from comparisons with the latest models like Funcodec [2], Encodec [3], and DAC [4]. Furthermore, the paper should include comparisons with the experimental results of DinoSR for both the speech recognition and tokenization tasks, as the encoder component of the paper is primarily based on the DinoSR model.

3. Lastly, the metrics used for evaluating downstream tasks are insufficient. For example, in the speech generation task, subjective evaluations such as Mean Opinion Score (MOS) should be included, as this is a critical metric. For speech tokenization, additional metrics related to speech reconstruction, such as Mel/STFT distance, could be incorporated.

[1] Ju, Zeqian, et al. "Naturalspeech 3: Zero-shot speech synthesis with factorized codec and diffusion models." arXiv preprint arXiv:2403.03100 (2024).

[2] Du, Zhihao, et al. "Funcodec: A fundamental, reproducible and integrable open-source toolkit for neural speech codec." ICASSP 2024-2024 IEEE International Conference on Acoustics, Speech and Signal Processing (ICASSP). IEEE, 2024.

[3] Défossez, Alexandre, et al. "High fidelity neural audio compression." arXiv preprint arXiv:2210.13438 (2022).

[4] Kumar, Rithesh, et al. "High-fidelity audio compression with improved rvqgan." Advances in Neural Information Processing Systems 36 (2024).

**Questions:**

1. How does the experimental section provide evidence for the claim that simultaneous pre-training on speech representation learning and generation enhances performance compared to pre-training on only one task, and what justifies the assertion of reduced overhead and cost?

2. Could you include comparisons with models like Natural Speech 3, Funcodec, Encodec, DAC, and DinoSR to improve the evaluation of the proposed methods in speech generation and tokenization tasks?

3. Could you incorporate additional subjective and objective metrics, such as Mean Opinion Score (MOS) and Mel/STFT distance, provide a more comprehensive assessment of model performance?

---

> ### Author Response · Authors · 2024-11-24
>
> > Firstly, I have some doubts regarding the motivation and novelty validation in this paper.  ...
>
> As clearly stated in the limitations, we acknowledge the fact that performance drop in recognition tasks compared to existing self-supervised learning models at similar scales is not ideal. Nevertheless, we believe this work is still valuable for different reasons:
>
> - Unified pre-training has shown great value in other application fields but has not yet been explored in speech. For example:
>   - In NLP, next-token prediction has become the predominant pre-training method. Back then there were different pre-training approaches for different purposes (such as contrastive learning for retrieval, masked language modeling for representation learning, etc.), much like the current research landscape of speech. The unified pre-training method has significantly boosted the progress of NLP in both understanding and generation tasks.
>   - In image processing, joint representation learning and generation have also been studied more recently [A]. Similar to our work, [A] found a trade-off between image classification accuracy and generation quality. However, their effort is still recognized and many later works followed the spirit to push the boundary of unified pre-training.
> - Unified pre-training also has its own advantage
>   - This is related to the question raised by the reviewer:
>    > If these advantages cannot be clearly demonstrated, then why combine the two tasks?
>
>     In certain tasks, having both good representation and generation are equally important. Our experiment on speech tokenization and resynthesis is an example. We show a clear gap between our method and other methods, where speech representation and generation are optimized separately.
>
> To summarize, we believe these are strong evidence to support the importance of the research direction this work is pursuing. Once again, we acknowledge the fact that this model is not state-of-the-art when competing against specialized models. Instead of chasing state-of-the-art through pre-training for either discriminative or generative tasks, we contribute to the field by developing a different way of pre-training that is more general. Therefore, we argued that our work, as the first step towards unifying pre-training for speech representation learning and generation, is well-motivated and novel.
>
> ---
>
> > Secondly, the selection of baselines in the paper is quite limited, especially for the speech generation and tokenization tasks. Speech generation could be compared with Natural Speech 3 [1]
>
> We thank the reviewer for suggesting more baseline models to compare against.
>
> For speech generation, we would like to point out that all state-of-the-art models are not open-sourced to the best of our knowledge, and we therefore needed to reproduce one baseline ourselves. We choose to reproduce SpeechFlow (instead of Natural Speech 3) for several reasons:
> - it also falls in the pretrain-and-finetune paradigm, using the same labeled and unlabeled datasets (that are both publicly available) as our work; Natural Speech 3 requires 60k hours of labeled data where the annotations are not publicly available.
> - Architecture-wise it is more comparable to the proposed work, whereas Natural Speech 3 has a 4-stage diffusion design with disentangled input feature
> - Following SpeechFlow’s evaluation protocol (continuing zero-shot TTS) allows us to compare directly with VALL-E and Voicebox without the need for model weights.
>
> > the speech tokenization task could benefit from comparisons with the latest models like Funcodec [2], Encodec [3], and DAC [4].
>
> For speech tokenization, we targeted SpeechTokenizer as it was the most recent work in this direction at the time. **We have added the earlier works suggested by the reviewer to Table 2, including Encodec and DAC.** We would like to note that, unlike the original models studied in Table 2, the models suggested by the reviewer are NOT developed for speech tokenization but for general audio compressing. Therefore they inevitably performed poorly on low-bit-rate speech-only tokenization,  especially on encoding spoken content (see WER in Table 2).
>
> >  Furthermore, the paper should include comparisons with the experimental results of DinoSR for both the speech recognition and tokenization tasks, as the encoder component of the paper is primarily based on the DinoSR model.
>
> For DinoSR, the original work focused on a 12-layer model only and did not provide a recipe for training a 24-layer model. We also note that we failed to train a 24-layer model DinoSR with the official code, bottlenecked by training stability. (we did not extensively sweep the hyper-parameters due to resource restriction)
> Nevertheless, **a comparison between DinoSR and UniWav can be found in Table 3 by comparing the fine-tuned WER with and without decoder during pre-training**. Note that DinoSR is modified here to taking Encodec feature as input.
>
> ---
>
> (To be continued)

---

> ### Author Response · Authors · 2024-11-24
>
> > Lastly, the metrics used for evaluating downstream tasks are insufficient. For example, in the speech generation task, subjective evaluations such as Mean Opinion Score (MOS) should be included, as this is a critical metric. For speech tokenization, additional metrics related to speech reconstruction, such as Mel/STFT distance, could be incorporated.
>
> We thank the review for these helpful suggestion, we have updated the paper accordingly:
> - For TTS, **we have added subjective evaluation results to Section A.1 and Table 6**. Result confirmed that our model is indeed matching the performance of SpeechFlow, which is a generative-only pre-training method.
> - For reconstruction, **we have added ViSQOL to Table 2**. ViSQOL is an objective and intrusive metric that measures audio reconstruction quality, similar to Mel/STFT distance but more widely adopted by codec models (including Encodec and DAC, which were added to Table2 as suggested by the reviewer).
>
> ---
>
> We hope the responses and updates on the paper have addressed the concerns raised by the reviewer.
>
>
> ### Reference
>
> [A]  https://arxiv.org/abs/2211.09117

---

> > ### Comment · Reviewer_8mpX · 2024-11-28
> >
> > Thanks to the authors for their response and the additional experiments. These have addressed some of my concerns, and I am willing to raise my score to 5. However, I still share a similar concern with Reviewer g67C. While I agree that joint pretraining of speech understanding and generation is a highly significant direction, I do not see the advantages of this approach clearly demonstrated in the paper. The authors should explain the benefits that joint pretraining of these two tasks can bring or provide insights into certain tasks to better demonstrate the effectiveness of UniWav.

---

### Official Review · Reviewer_g67C · 2024-11-04

**Soundness:** 3
**Presentation:** 3
**Contribution:** 3
**Rating:** 5
**Confidence:** 3

**Summary:**

UniWav is an encoder-decoder framework designed to unify pre-training representation learning and generative tasks. With the appropriate design choices for pre-training, UniWav can jointly learn a representation encoder and generative audio decoder that can be applied to both types of tasks.

**Strengths:**

The paper introduces the first unified pre-training framework (UniWav) for speech representation learning and generation.
UniWav can compete with different foundation models with low bitrate speech tokenization and high-quality resynthesis.

**Weaknesses:**

1. The results of the method in this paper do not show an advantage over existing methods in speech recognition and speech generation.
2. In the speech tokenization section, there is a lack of experiments related to the modeling performance of the speech language model (LLM-TTS) as mentioned in SpeechTokenizer[1]. Such experiments could effectively evaluate the potential of the tokenizer proposed in this paper when applied to autoregressive speech synthesis.
[1]

**Questions:**

Can the method proposed in this paper be applied to general audio such as sound and music?

---

> ### Author Response · Authors · 2024-11-24
>
> > The results of the method in this paper do not show an advantage over existing methods in speech recognition and speech generation.
>
> As clearly stated in the limitations, we acknowledge the fact that performance drop in recognition tasks compared to existing self-supervised learning models at similar scales is not ideal. Nevertheless, we believe this work is still valuable for different reasons:
>
> - Unified pre-training has shown great value in other application fields but have not yet been explored in speech. For example:
>   - In NLP, next-token prediction has become the predominant pre-training method. Back then there were different pre-training approaches for different purposes (such as contrastive learning for retrieval, masked language modeling for representation learning, etc.), much like the current research landscape of speech. The unified pre-training method has significantly boosted the progress of NLP in both understanding and generation tasks.
>   - In image processing, joint representation learning and generation have also been studied more recently [A]. Similar to our work, [A] found a trade-off between image classification accuracy and generation quality. However, their effort is still recognized and many later works followed the spirit to push the boundary of unified pre-training.
> - Unified pre-training also has its own advantage.
>   - Having both good representation and generation are equally important in certain tasks.
>   - Our experiment on speech tokenization and resynthesis is an example. We show a clear gap between our method and other methods, where speech representation and generation are optimized separately.
>
> To summarize, we believe these are strong evidence to support the importance of the research direction this work is pursuing. Once again, we acknowledge the fact that this model is not state-of-the-art when competing against specialized models. Instead of chasing state-of-the-art through pre-training for either discriminative or generative tasks, we contribute to the field by developing a different way of pre-training that is more general. Therefore, we argued that our work, as the first step towards unifying pre-training for speech representation learning and generation, is well-motivated and novel.
>
> ---
>
> > In the speech tokenization section, there is a lack of experiments related to the modeling performance of the speech language model (LLM-TTS) as mentioned in SpeechTokenizer[1]. Such experiments could effectively evaluate the potential of the tokenizer proposed in this paper when applied to autoregressive speech synthesis. [1]
>
> We thank the reviewer for the helpful comment, and we will add this to our future work. We also note that it is hard to provide an apple-to-apple comparison between different codec models in training token-based auto-regressive TTS, since there are different in many aspects:
> - Different bitrate results in different TTS configuration. E.g., [1] required modeling 400 tokens per second, whereas our method required 50 or 100 tokens per second.
> - For RVQ-based codec TTS such as [1,B,C], training 2 models (typically an auto-regressive model for semantic codes and a non-auto-regressive model for remaining codes) for text-to-speech is often necessary to deal with multiple codebook. In this case these TTS system are not fully auto-regressive.
>
>
> ---
>
> > Can the method proposed in this paper be applied to general audio such as sound and music?
>
> We believe the generation part of UniWav can be easily applied to sound and music, as flow matching have been explored in the [D]. The representation learning part should also be applicable to sound and music. However, the input feature choice and the masking strategy might need to be revisited to adapt to those different domains [E]. We thank the reviewer for pointing out an important future direction, we have updated Section 5 to include the idea.
>
> ---
>
> ### Reference
>
> [A] https://arxiv.org/abs/2211.09117
>
> [B] https://arxiv.org/abs/2410.00037
>
> [C] https://arxiv.org/abs/2301.02111
>
> [D] https://arxiv.org/abs/2312.15821
>
> [E] https://arxiv.org/abs/2110.09784

---

> ### Comment · Reviewer_g67C · 2024-11-27
> **Official Response by Reviewers**
>
> I believe that the joint pretraining of audio understanding and generation is a promising and exciting research direction. I appreciate the authors' attempts in this area. However, my major concern is that the pretraining framework proposed by the authors does not show significant advancements over previous methods (like TTS, ASR, and tokenizers). Therefore, I have doubts about whether the framework is an effective representation learning method.
>
> - ASR: For the ASR task, the proposed method does not show advantages over Data2Vec 2 and even HuBERT.
>
> - TTS: Regarding the TTS task itself, the current state-of-the-art methods are NaturalSpeech2/MASKGCT and CosyVoice/FishSpeech (a non-autoregressive and an autoregressive model). The paper does not include these baselines for comparison.
>
> - TTS Tokenizer: For the tokenizer, the current state-of-the-art open-source tokenizers for TTS are CosyVoice and DAC. The authors also do not include these as baselines. Moreover, for the TTS tokenizer, the frame rate (how many tokens per second) is more important than the bitrate for LLM-based CodeTTS models, which could be included in Table 2 for comparison. Additionally, recent low-bitrate codec models such as WavTokenizer and Mini are also not included in the paper for comparison.
>
> Indeed, I want to emphasize that I do not believe a unified model must achieve optimal results across all aspects, especially when the scale is not significantly increased. It would be more beneficial to demonstrate some potential synergies. For instance, combined training for understanding and generation could significantly assist each other, or the trained tokenizer might possess properties that other training methods in semantic and acoustic processing do not have. However, based on the scores presented in the paper, the performance across various metrics appears to be somewhat unsatisfactory.

---

### Meta-Review · Area_Chair_sxq4 · 2024-12-08

**Metareview:**

This paper introduces a novel speech self-supervised learning (SSL) approach by combining existing methods DinoSR and SpeechFlow. This hybrid method shows strong performance on both generative and discriminative tasks, outperforming models that specialize in only one type. The paper conducts extensive experiments across various tasks, including speech recognition, text-to-speech, and tokenization. It also includes detailed analyses like ablation studies and mutual information metrics. The paper is well-organized and clearly presents technical details, such as the encoder-decoder structure and the Flow Matching method. Visual aids and concise explanations enhance the understanding of complex concepts. The work is expected to inspire future research in the speech community, given its innovative approach and promising results. It shows SOTA performance on generative tasks and competitive results on discriminative tasks, particularly in speech tokenization.

**Additional Comments On Reviewer Discussion:**

The reviewers raised concerns 1)  The method does not show an advantage over existing methods in speech recognition and generation; 2) Missing experiments on the modeling performance of the speech language model (LLM-TTS) in the speech tokenization section; 3)  The paper lacks evidence supporting the claimed benefits of combining speech representation learning and generation, and the results do not strongly validate the stated motivations; 4)  Limited selection of baselines, especially for speech generation and tokenization tasks; 5) Insufficient metrics for evaluating downstream tasks, such as missing subjective evaluations like MOS for speech generation; 6) Evaluations are only conducted on Librispeech, which has homogeneous speaker conditions, lacking robustness across different domains; 7) The paper could benefit from additional experiments and discussions on the modeling choices, such as the impact of fine-tuning the encoder for ASR; 8) The paper does not adequately address the performance of UniWav on speaker recognition tasks; 9) Lack of results for low-resource scenarios and using a frozen encoder, which are essential to support the claims about the encoder's effectiveness. Most of them have been addressed during the author-reviewer discussion.

---

### Decision · Program_Chairs · 2025-01-22

Accept (Poster)